# Image Caption Generation via Unified Retrieval and Generation-Based Method

**Shanshan Zhao** [1,2,†,‡], **Lixiang Li** [1,2,*,‡], **Haipeng Peng** [1,2,‡] , **Zihang Yang** [1,2,‡] **and Jiaxuan Zhang** [1,2,‡]

1. Information Security Center, State Key Laboratory of Networking and Switching Technology, Beijing University of Posts and Telecommunications, Beijing 100876, China; shanshan_zhao@bupt.edu.cn (S.Z.); penghaipeng@bupt.edu.cn (H.P.); 2019110960@bupt.edu.cn (Z.Y.); zhangjiaxuan@bupt.edu.cn (J.Z.)
2. National Engineering Laboratory for Disaster Backup and Recovery, Beijing University of Posts and Telecommunications, Beijing 100876, China
* Correspondence: lixiang@bupt.edu.cn; Tel.: +86-010-6228-2264
† Current address: Beijing University of Posts and Telecommunications, Haidian District, P.O. Box 145, Beijing 100876, China.
‡ These authors contributed equally to this work.

**Abstract:** Image captioning is a multi-modal transduction task, translating the source image into the target language. Numerous dominant approaches primarily employed the generation-based or the retrieval-based method. These two kinds of frameworks have their advantages and disadvantages. In this work, we make the best of their respective advantages. We adopt the retrieval-based approach to search the visually similar image and their corresponding captions for each queried image in the MSCOCO data set. Based on the retrieved similar sequences and the visual features of the queried image, the proposed de-noising module yielded a set of attended textual features which brought additional textual information for the generation-based model. Finally, the decoder makes use of not only the visual features but also the textual features to generate the output descriptions. Additionally, the incorporated visual encoder and the de-noising module can be applied as a preprocessing component for the decoder-based attention mechanisms. We evaluate the proposed method on the MSCOCO benchmark data set. Extensive experiment yields state-of-the-art performance, and the incorporated module raises the baseline models in terms of almost all the evaluation metrics.

**Keywords:** image caption; computer vision; natural language processing

## 1. Introduction

Recently, the multidisciplinary image captioning task has received considerable critical attention in natural language processing and computer vision. It aims to generate human-like descriptions according to the input image. It is a very difficult and challenging task. Since not only must the caption generation models to recognize the objects and their relationships in an image, but also must expressing them in form of natural language. Apparently, the input image is always represented in the vision domain, while the output sentence is in another irrespective language domain. Despite the difficulties of the cross-domain operation, more important thing is that it has a wide range of applications, such as text-based image retrieval, human-robot interaction, etc.

Most of the existing captioning models can be categorized into template-based, generation-based and retrieval-based methods. The retrieval-based method first retrieves the visually similar images with their captions from the training data set based on the visual similarity of image features, and these existing captions are called candidate captions [1]. For each query image, the retrieval-based method

selects the most appropriate caption from the candidate captions. For example, Ordonez et al. [2] builds a web-scale collection of images with associated descriptions from the Internet. They use the gist feature and the resized tiny image feature respectively, to retrieve the similar images and associated descriptions. Their final result was the captions from the top ranked image. Gupta et al. [3] finds *k* images that were most similar to the query from the training images, and used the phrases extracted from their descriptions. A ranked list of triples was generated, and they were used to compose the descriptions for the new image. Kuznetsova et al. [4] presented a tree-based approach to compose expressive image descriptions that makes use of naturally occurring web images with captions. The tree fragments were expressive phrase from existing descriptions. Then they composed a new description by selectively combining the extracted tree fragments. These methods can produce general and syntactically correct captions. However, the generated caption may not be semantic suitable and novel for the query image.

Inspired by the sequence-to-sequence neural machine translation model, most of the successful image captioning generation-based methods adopt the encoder–decoder framework and they are trained in an end-to-end manner. Generally, the convolutional neural network (CNN) acts as the encoder to extract the visual features of the input image, and the recurrent neural network (RNN) is used as a decoder to translate the input visual feature vectors into natural captions word by word. However, these methods are unable to analyze the image over time during the caption generation process. As a result, the attention mechanism [5–7] establishes the connection of the caption words and their related image regions dynamically during the decoding process [8–10], which highly improved the captioning performance. Despite better quality and flexibility of the generation-based results, the generated captions mainly suffer from a lack of fluency, diversity and informativeness [11].

In this paper, we work toward constructing a framework that takes advantage of the generation-based method and the retrieval-based method. First, we use the retrieval-based method to search the most similar image with their related descriptions of the query image from the MSCOCO [12] training data set. Secondly, for the extracted visual features of the query image, we fully explore potential hidden correlations of the different image regions to form a spatial and relational representation. Our de-noising module models the cross-modal correlations of the attended visual features from vision domain and the embedded similar sequences from language domain. Finally, the decoder takes both the de-noised semantic knowledge and the image representations to generate the final output descriptions. In general, our main contributions of this work can be summarized as follows:

- We propose a model that incorporate the retrieval-based and the generation-based image captioning method to translate the source image into target language.
- The proposed de-noising module incorporated with the visual encoder can be viewed as a preprocessing component, providing a part of high-level semantic information and a set of attended visual features.
- This component can be used with the decoder-based attention mechanisms. Experiment result shows the effectiveness when we replace the information sources for different baselines.
- The experiment result evaluated on the MSCOCO data set shows that our model has achieved the state-of-the-art results.

## 2. Related Works

### 2.1. Retrieval-Based Methods

The retrieval-based conventional methods can retrieve the visually similar images and their captions from the training data set. This kind of approach can directly copy sentences from these retrieved captions. Mason et al. [13] uses GIST features of the query image to select the nearest image and sentence. The related sentences are compressed by removing details that may probably not semantic suitable for the test image. Devlin et al. [14] chose one caption from the retrieved

captions pool of $k$ nearest training images. The captions of the query image are then used to select the most appropriate captions from the candidate pool. However, these methods heavily rely on the pre-constructed candidate pool, and the selected captions may not be visual and semantic correct for the query image. Xu et al. [11] propose a framework by using adversarial learning, and leverage the copying mechanism to extract words from the retrieved captions to enrich the meaning of the generated captions. They got the competitive experiment result compared with the baselines.

*2.2. Generation-Based Methods*

Inspired by the success of the encoder–decoder framework used in deep neural machine translation tasks [15,16], the image captioning tasks adopted the structure to translate the source image into the target language [17–19]. Vinyals et al. [20] first applied the framework into the image captioning task, which trained a convolutional neural network(CNN) as the encoder, used these extracted visual features as the input to a recurrent neural network(RNN) decoder, and generated the captions word by word until getting the end token of the sentence. These encoder–decoder-based models [21,22] gained promising performance.

Furthermore, the attention mechanism was introduced to the caption generation. Xu et al. [5] used the deterministic soft attention and the stochastic hard attention to help the decoder focus on the most relevant image part when generating different words, and they promoted the sentence generation quality. Anderson et al. [23] further improved the attention module by the bottom-up and top-down mechanism. Attention mechanism bridged the gap between the vision domain and the language domain effectively. Thus, this mechanism has been widely adopted in the image captioning tasks [6]. Vaswani et al. [7] proposed a *Transformer* model which stacked the attention blocks entirely without convolution and recurrence. This model composes an encoder and a decoder. The encoder comprised a self-attention and a position-wise feed-forward block. The decoder contained a self-attention and a cross-attention layer. Zhu et al. [24] adapted their framework from the *Transformer* architecture to perform image captioning. The difference lies in that they replace the encoder of the *Transformer* by a CNN model. Herdade et al. [25] incorporates the information about the spatial relationships of the R-CNN detected object pairs to the attention block. Huang et al. [26] proposed an "Attention on Attention" module to model the relationships of different objects of the image on encoder, and refining for the decoder with self-attention mechanism. Pan et al. [27] propose a unified X-Linear attention block that fully leverage bilinear pooling to selectively capitalize on different visual information. Cornia et al. [28] propose a Meshed *Transformer* with memory for image captioning. The framework uses a mesh-like connectivity at decoding stage to exploit low- and high-level features.

Different with the above-mentioned methods, we design a novel image captioning model that incorporate the retrieval-based and the generation-based method which contains a visual encoder, a de-noising module and a decoder. The de-noising module does help to filter the retrieved similar sequences conditioned on the visual features. In addition, the decoder incorporates the visual features and the filtered textual features to decode into fluent descriptions.

## 3. Model Architecture

Most of the generation-based image to language translation models makes use of the encoder–decoder structure. Visual features are widely used as the source information for image captioning. For the input image $I$, an intuitive way to extract visual features is by ResNet [29], VGG [30] or GoogLeNet [31]. The encoder is usually used to encode image features, mapping the image $I$ to a set of visual features. The decoder uses these visual features to generate the output sequence. In this section, we briefly review the architecture of the proposed model. The proposed method is based on the attention mechanism. As can be seen in Figure 1, our model incorporates the retrieval-based method and the generation-based method. The retrieval-based method search the most similar image and their descriptions in the training data set for the input image. At the generation-based branch, a CNN is used to extract the visual features of the input image. These visual

features are then fed into the visual encoder to compute their inner relationships. The de-noising module filtered the embedded similar sequences conditioned on the attended visual features yielded a set of textual features. A caption decoder leverages both the attended visual features and the de-noised semantic features to generate the output descriptions.

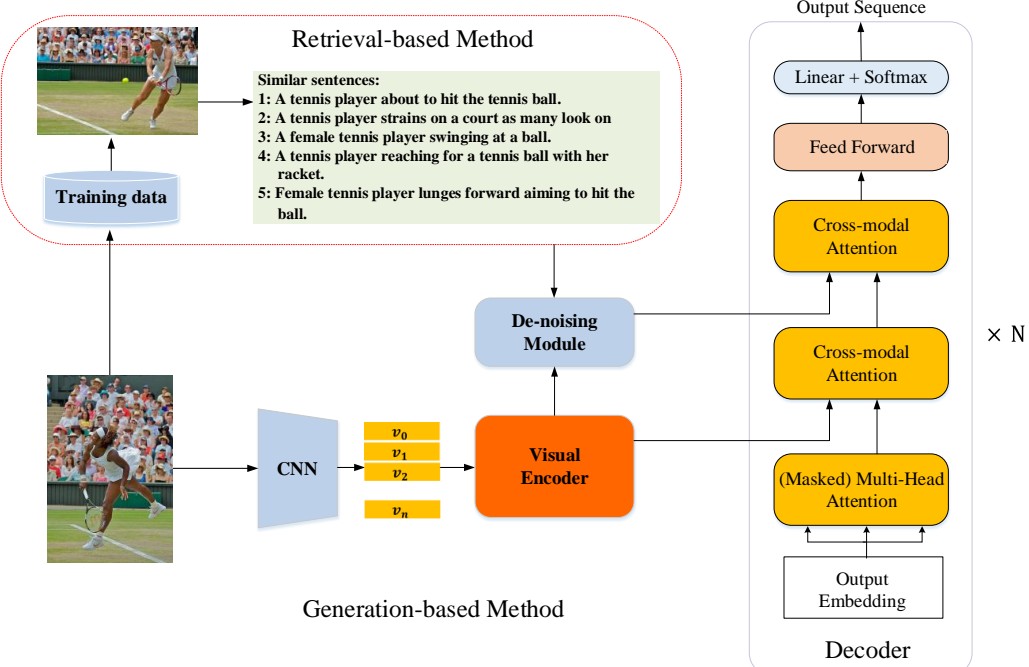

**Figure 1.** Our model combined the retrieval-based method and the generation-based method. The de-noising module here is used to filter the retrieved similar sequences. The decoder here used both the filtered textual features and the attended visual features to generate the final descriptions.

### 3.1. Preliminary

The *Transformer* model has been widely applied in sequence-to-sequence tasks and gains significant improvements. Our model mainly acquires the attention mechanism proposed by [7]. Thus, we first introduce the preliminary knowledge of the scaled-dot product attention which is defined as:

$$Att(Q,K,V) = softmax(\frac{QK^T}{\sqrt{d_k}})V \tag{1}$$

where $Q \in R^{m \times d_{model}}$ is the query matrix, $m$ is the number of objects, $d_{model}$ is the latent dimension. $K \in R^{n \times d_{model}}$ denotes the key matrix, and $V \in R^{n \times d_{model}}$ represents the value matrix. $K$ and $V$ are key-value matrix pairs. In briefly, the query matrix $K = [k_1, ..., k_n]$, $k_n \in R^{d_{model}}$, the value matrix $V = [v_1, ..., v_n]$, $v_n \in R^{d_{model}}$, and $n$ is the number of the key-value pairs. $QK^T$ stands for the product operation to compute the similarity score between $Q$ and $K$, which may cause the result become larger or smaller and further influence the precision of the variable. Thus, $\sqrt{d_k}$ indicates the scaled dot to scale $QK^T$. The attention of queries $Q = [q_1, ..., q_m]$, $q_m \in R^{d_{model}}$ is computed in parallel. After a series computation, $Att(Q,K,V) \in R^{m \times d_{model}}$ is the attended output features.

The multi-head attention contains $h$ parallel heads, and each head corresponds to a scaled-dot-product attention function. These $h$ scaled-dot product attention are independent. The multi-head attention concatenated them and projected to form the final attentive representation:

$$head_i = Att(QW_i^Q, KW_i^K, VW_i^V) \tag{2}$$

$$O = MultiHead(Q,K,V) = Concat(head_1, ..., head_h)W^o \tag{3}$$

where $Concat(\cdot)$ is the concatenation function, $head_i$ represents the $i$-th scaled-dot-product attention. $W^o \in R^{h*d_v \times d_{model}}$ is the output of different heads, $d_v$ is the output dimension of each head. $W_i^Q \in R^{d_{model} \times d_k}$, $W_i^K \in R^{d_{model} \times d_k}$, $W_i^V \in R^{d_{model} \times d_v}$ are the linear *Transformer* parameter matrices.

Followed the multi-head attention is a position-wise feed-forward network(FFN), FFN takes the output of the multi-head attention as input which is defined as:

$$FFN(x) = Dropout(max(0, xL_f + b_f))L_{ff} + b_{ff} \qquad (4)$$

where $Dropout(\cdot)$ is the dropout [32]operation to prevent over-fitting, $max(\cdot)$ function is the ReLU activation, $L_f$ and $L_{ff}$ represent two linear transform operations, and $b_f$ and $b_{ff}$ are the corresponding bias terms. It is worth noting that both the multi-head attention and the FFN all followed a series operation of residual connection, dropout and layer normalization [33].

### 3.2. Visual Encoder

When we look at an image, we always focus on a specific object first. Then we look around to find the other semantically or spatially related objects. These related objects form inner groups that are of importance to our approach. The visual encoder here is used to learn the spatial or semantic relationships of these objects. Our visual encoder is composed by a multi-head self-attention and a position-wise feed-forward network as can be seen in the left of Figure 2. The input of the self-attention sub-layer is a set of visual features of the image.

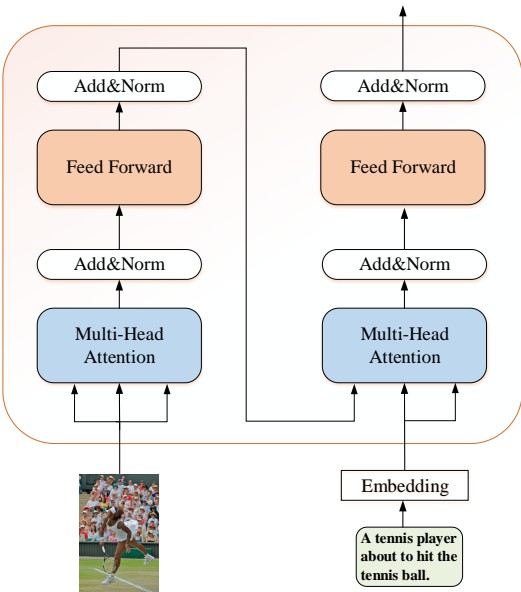

**Figure 2.** The left part is a visual encoder, the right part is the de-noising module. Here we list one of the retrieved similar sequences.

Given a source image $I$, we use the top $m$ objects extracted by Faster R-CNN [34] to represent the image, and each object is represented as a feature vector $x_i \in R^{d_x}$. The image includes $m$ objects can be denoted as a visual feature matrix $X = (x_1, ..., x_m) \in R^{m \times d_x}$. We adapt the dimensionality of the visual feature by using the fully connected layer to fit the encoder. The linear projected visual feature matrix $X \in m \times d_{model}$ is fed into the multi-head attention to mimic our human how these visual features $(x_1, x_2, ..., x_m)$ are related. Multi-head attention block composes $h$ parallel heads.

The full visual encoder computation process can be described as an iteration:

$$V_1 = FFN(MultiHead(X, X, X)) \qquad (5)$$

$$V_N = FFN(MultiHead(V_{N-1}, V_{N-1}, V_{N-1})) \qquad (6)$$

where $X$ is a set of extracted original input visual features. Here we use $d_k = d_v = 64$. $V_1$ is the nonlinearly transformed attended features of the first iteration. After we obtain $V_1$, the first iteration is completed. The second iteration takes the computation result of the first time as input and repeats the same process for $N$ times to get the final output by using Equation (6). In this paper, we denote the final attended visual feature $V_N$ of the visual encoder as $V$. Followed with the self-attention layer and the FFN follows a residual connection and a layer normalization post-processed operation to normalize the visual features, respectively.

### 3.3. De-Noising Module

To make use of the retrieved similar sequences for image captioning, we must filter these sequences conditioned on the attended visual features. As can be seen in the right bottom of Figure 2, a retrieved sequences seems semantically similar to the original input image which can provide complementary semantic information. The first cross-modal attention sub-layer can learn the correlated features by using the attended visual features $V$ to query the textual features $T$. We leverage $V$ as query, $T$ as the key-value pairs. By using the multi-head attention operation, we get a set of the textual features that most related with the visual features. The text encoder can be denoted as follows:

$$T_1 = FFN(MultiHead(V, T, T)) \tag{7}$$

$$T_N = FFN(MultiHead(V_N, T_{N-1}, T_{N-1})) \tag{8}$$

where $V \in R^{m \times d_{model}}$, and $T \in R^{l \times d_{model}}$. $T_1$ is the textual features embedded by the embedding model. The query matrix $V$ is from the vision domain, and $T$ is the key-value pairs from the language domain. At last, the filtered textual features only contain homogenous information since the information from the vision domain only serves as the attentive weight and is not part of the final values.

### 3.4. Decoder

Given the attended visual features $V$ and the filtered textual features $T$, the decoder will make full use of this information to generate descriptions in a recursively manner for the image. We can see in Figure 1 inside the decoder, there is a masked self-attention block, two multi-head cross-modal attention components and a position-wise feed-forward layer.

The input of the masked self-attention block is a temporal input caption which can be embedded into a feature matrix $Y = [y_1, ..., y_n] \in R^{n \times d_{model}}$, and $n$ is the input captions length. Considering that the prediction of a word should only use the previous predicted words, which means that we should prevent the decoder from seeing the future sequence information. When we generate the $t$-th word, we only require the former predicted words information $Y_{\leq t}$. We implement this by setting the subsequent sequence position to $-\infty$. Therefore, a triangular masked matrix with dimension $M \in R^{n \times n}$ is generated as the input to the first sub-layer, and $n$ is the length of the input caption. This first sub-layer models the self-attention on the caption words which can be described as:

$$M1 = MultiHead(Y, Y, Y) \tag{9}$$

where $Y \in R^{n \times d_{model}}$ is the masked caption feature. The output of the first sub-layer $M1$ are then fed into the second cross-modal attention sub-layer as the query. We use the attended visual feature $V$ as the key-value pairs.

The second cross-modal attention layer can be defined as follows:

$$M2 = MultiHead(M1, V, V) \tag{10}$$

where $M1 \in R^{n \times d_{model}}$ is the textual feature, and $V \in R^{m \times d_{model}}$ is the attended visual feature from the visual encoder. $M1$ is the query, and $V$ is the key-value pairs. $M2$ is a set of attended visual features.

For the third multi-head attention sub-layer, we adopt the de-noised text features $T^{l \times d_{model}}$ as the key-value pairs, $M_2$ as the query. In addition, we have:

$$M3 = MultiHead(M2, T, T) \qquad (11)$$

where $M2 \in R^{n \times d_{model}}$. Finally, $M3 \in R^{n \times d_{model}}$ is fed into the position-wise feed-forward network to nonlinearly transform this feature :

$$F = FFN(M3) \qquad (12)$$

After the above calculation, a linear word embedding layer takes $F$ as input to transform it into a $d_v$-dimensional space, where $d_v$ is the vocabulary size. Subsequently, the softmax function is performed on each word to predict the probability over words in the dictionary. The decoder contains $N$ layers, thus the decoder repeats the same process for $N$ times.

### 3.5. Training Details

We first train the proposed model by the word-level cross-entropy loss(XE). Following the common practice in [8], our model predicts the next token according to the previous ground-truth words which are not the predicted words. The cross-entropy loss $L_{XE}$ is defined as:

$$L_{XE}(\theta) = - \sum_{t=1}^{T} log(p_\theta(y_t^* | y_{1:t-1}^*)) \qquad (13)$$

where $t$ is the $t$-th word, and $y_{1:T}^*$ represent the target ground-truth sequence.

Then we train the proposed model by reinforcement learning to optimize the non-differentiable evaluation metrics CIDEr-D which is close to our human judgment. In the decoder phase, we use beam search [23] to sample top-$k$ words from the decoder probability distribution, and keep the top-$k$ highest probability sequences. We adopt the self-critical sequence training method on these sampled sequences which is:

$$\nabla_\theta L_{RL}(\theta) = - \frac{1}{k} \sum_{i=1}^{k} ((r(y_{1:T}^i) - b) \nabla_\theta log p(y_{1:T}^i)) \qquad (14)$$

where $y_{1:T}^i$ is the $i$-th sentence in the beam, $r(\cdot)$ is the reward function uses the score of CIDEr-D, and $b = r(y_{1:T})$ is the reward computed on the greedy decoding sequences.

## 4. Experiments

To take advantage of the generation-based and the retrieval-based method, we must retrieve the visually similar images for each queried image in the MSCOCO data set. For this purpose, we use Baidu-API to search for the most similar image in the data set for each image. Then we can acquire the corresponding five ground-truth descriptions of the similar image. After the above operation, we can use the de-noising module to get a set of attended textual features conditioned on the visual features.

For the ground-truth captions and the retrieved similar sequences, we count the times of each word that appears in the sentences and replace the words that appear less than five times by the unknown word token. The rest words are used to construct the dictionary with 9487 words. Each word is embedded into $d_{model}$ vectors. If the length of the sequences is shorter than 16 words, we will use zero-padding to fill them to the maximum size.

In this section, we evaluate our method on the MSCOCO image captioning data set. We make comparisons with several state-of-the-art methods. Furthermore, we integrate the de-noising module with the visual encoder as a preprocessing component and added to the decoder-based models. The models takes both the attended visual features and the textual features as input.

### 4.1. Datasets and Metrics

MSCOCO is a popular benchmark data set and it has been widely used in the tasks like object detection, image captioning and instance segmentation. This data set contains 82,783 training images, and 40,504 validation images. To make fair comparisons with other methods, we use the commonly used data set splits provided by [35] to evaluate our offline evaluation result. The re-split training set contains 113,287 training images, 5000 validation images and 5000 test images respectively, and each image is paired with 5 annotated captions. The result of the proposed model is reported on the Karpathy split testing data set.

To evaluate the generated caption quality, we use the official MSCOCO image captioning evaluation metrics CIDEr [36], SPICE [37], BLEU [38], METEOR [39] and ROUGE [40] that are widely used in previous works. CIDEr measures the consensus in the task by using Term Frequency-Inverse Document Frequency weighting for each $n - gram$. BLEU is the precision metric by measuring the similarity of the generated captions and the ground-truth captions. METEOR is a metric to explicit the word to word matches of the generated captions and the ground-truth captions [24]. SPICE uses the graph-based image semantic to compute F-score value of the objects, attributes and relationships in the generated caption. ROUGE is the metric that computes the longest common sub-sequence between the generated captions and the ground-truth captions, and the longer the common sub-sequence, the higher the score.

### 4.2. Experimental Settings

For our captioning model, we first use Faster R-CNN with ResNet-101 [29] as the feature extractor which is finetuned on the Visual Genome data set. We keep top $m$ objects to represent an image, and each object is represented by a 2048-dimensional feature vector $d_x$. In our model, we set $m = l = 17$. The mini-batch size is set to 16. The dropout rate is 0.9 which is used after the attention block and the position-wise feed-forward layer. Same with the base *Transformer* model, we set the dimension $d_{ff}$ of the feed-forward network to 2048. The latent dimension $d_{model}$ in the multi-head attention module is 512, the number of head $h$ is 8, the dimension of each head is $d_h = 64$, and the number of layers $N$ is 6.

To train the proposed model, we use the Adam method [41] with mini-batch 16 to update the parameters. We use the cross-entropy loss first to train the model for 10 epochs. The learning rate is set to $5e - 4$. Then we further train the proposed model for additional training by using the self-critical loss to directly optimize the CIDEr metric. The learning rate is $1e - 5$. The self-critical sequence training brought dramatic improvement of all the evaluation metrics. We set the beam size equals 4 to add the sequence diversity.

When we integrate the visual encoder and the de-noising module as a preprocessed component. After the self-attention computation of the visual encoder, the attended visual features $V \in d^{m \times d_{model}}$ is generated. We can simply add V with the de-noised textual features $T$ and followed with layer normalization. Then we fed the added sum result into the decoder and make comparisons with the original baseline models to verify the useful of the component.

### 4.3. Comparison with the State-of-the-Art Methods

Our captioning model make comparison with several powerful state-of-the-art methods. The performance of these model are shows in Table 1. These models are the generation-based methods. VS-LSTM [42] replace the low-level visual features with semantic attributes. Bottom-Up and Top-Down Attention(Up-Down) [23] adopt the Faster R-CNN features as the encoder, and uses a two-layer LSTM as the decoder. ADP-ATT [6] learnt an adaptive attention model that automatically determined when to look and where to look for word generation. GCN-LSTM [43] used the Graph Convolutional Neural Network to exploit relationships of each two different entities in the image to perform image captioning. LSTM-A [44] integrated semantic attributes into the CNN plus LSTM framework for caption generation. SCST [8] makes use of attention on the grid of features and a one-layer LSTM as the

decoder. RFNet [45] exploited multiple CNNs as encoder and followed a recurrent fusion procedure to form representations to the decoder. UGRIC [11] is a model that incorporate the retrieval-based method and the generation-based method.

**Table 1.** Performance of our model and several state-of-the-art single models evaluated on MSCOCO "Karpathy" offline test split.

| Method | Cross-Entropy Loss | | | | | | Self-Critical Loss | | | | | |
|--------|------|------|------|------|-------|------|------|------|------|------|-------|------|
| | B@1 | B@4 | M | R | C | S | B@1 | B@4 | M | R | C | S |
| SCST [8] | - | 30.0 | 25.9 | 53.4 | - | - | 99.4 | - | 34.2 | 26.7 | 114.0 | 55.7 |
| ADP-ATT [6] | 74.2 | 33.2 | 26.6 | - | 108.5 | - | - | - | - | - | - | - |
| LSTM-A [44] | 75.4 | 35.2 | 26.9 | 55.8 | 108.8 | 20.0 | 78.6 | 35.5 | 27.3 | 56.8 | 118.3 | 20.8 |
| VS-LSTM [42] | 76.3 | 34.3 | 26.9 | - | 110.2 | - | 78.9 | 36.3 | 27.3 | - | 120.8 | - |
| Up-Down [23] | 77.2 | 36.2 | 27.0 | 56.4 | 113.5 | 20.3 | 79.8 | 36.3 | 27.7 | 56.9 | 120.1 | 21.4 |
| RFNet [45] | 76.4 | 35.8 | 27.4 | 56.5 | 112.5 | 20.5 | 79.1 | 36.5 | 27.7 | 57.3 | 121.9 | 21.2 |
| UGRIC [11] | - | - | - | - | - | - | 81.3 | 38.2 | 28.4 | 58.6 | 123.5 | - |
| GCN-LSTM [43] | 76.5 | 36.8 | 27.9 | 57.0 | 116.3 | 20.9 | 80.5 | 38.2 | 28.5 | 58.3 | 127.6 | 22.0 |
| Ours | 76.7 | 36.5 | 28.0 | 56.8 | 115.1 | 21.2 | 80.0 | 38.3 | 28.7 | 58.5 | 128.0 | 22.1 |

For fair comparison, we evaluate our model on the MSCOCO "Karpathy" test split. In Table 1, B@1, B@4, M, R, C and S correspond to the BLEU@1, BLEU@4, METEOR, ROUGE, CIDEr and SPICE scores. All values are reported as percentage. We show the single-model experiment results of our approach along with several competitive methods. As can be observed from Table 1 that our approach is superior to the first six models in all the evaluation metrics. After the model trained by the self-critical loss, the experiment results is a little worse on BLEU-1 of the UGRIC model, but our model is superior in terms of the other five evaluation metrics. We can see UGRIC unified the retrieval and generation-based method got 123.5 CIDEr score which is inferior to ours. Although GCN-LSTM is a very strong baseline. Our model is superior in terms of all the evaluation metrics except for Bleu-1 after self-critical sequence training. It can be seen that our model got 128.0 CIDEr score which is higher than GCN-LSTM. This indicates the competitive of our model. Although our model is superior to the baselines. It is worth mentioning that our model is inferior to the models [26–28].

In Figure 3 we show several examples selected from the validation set of the description results generated by our approach. The first column is the input image, the second column is the corresponding retrieved similar image. We can see that the retrieved result is similar to the input image. The first row of the original picture is describing two men fishing on a boat with a dog. The retrieved similar image is describing a man with a dog on the boat without the fishing action which is different with the input image. Although they are different in some detailed place, these similar sequences still bring useful textual information("boat", "dog"). The third row of the input image is a baby holding a toothbrush. The retrieved image and sentences are both visually and semantically similar to the input image. Our generated description is almost same with the ground truth. On the whole, the proposed method can generate fluency and semantic correct captions with help of the similar semantic information.

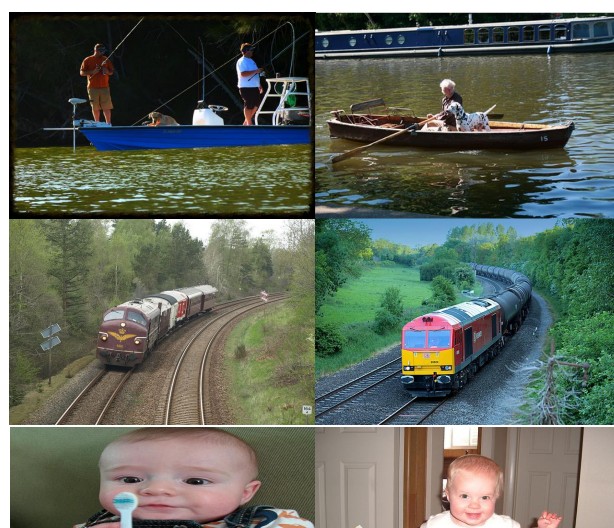

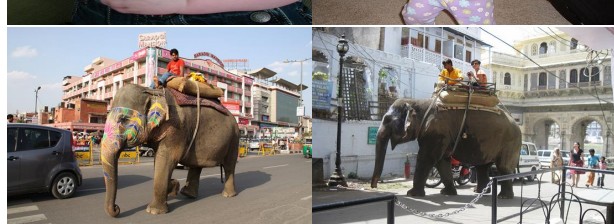

Retrieved: A person on a rowboat with a dalmatian dog on the boat.
Ours: Two men fishing on a boat with a dog on it
GT: Two men on a blue boat with dog fishing.

Retrieved: A train that is coming around the curb of a tree filled area.
Ours: A train is coming through tracks around with trees.
GT: A train is moving along train tracks through the woods.

Retrieved: A baby holding a toothbrush while smiling at the camera.
Ours: A baby holding a toothbrush in its hand.
GT: A child holds a toothbrush in their hand.

Retrieved: Elephants walking through a field on the side of road.
Ours: A man riding on the back of an elephant down a street
GT: A man is riding on top of an elephant.

**Figure 3.** Examples of the retrieved similar sequence and the captions generated by our model on MSCOCO validation set. "Retrieved" indicates one of the retrieved five similar sequences by computing the image pairs visual similarity. "Ours" is the generated result by our approach. "GT" denotes one ground-truth caption.

### 4.4. Ablation Study

Experiment results of different information sources for baselines. To verify the effectiveness of the de-noised textual features and the attended visual features, we replace the information sources of the following frameworks by the two kind features, respectively. The attended visual features is the output of the visual encoder. The added sum features is the summarization of the attended visual features and the de-noised textual features. The base models we compared with are as follows. Google NIC [46], ADP-ATT and Up-Down are three baselines that count on the visual features to generate captions. Google NIC uses the GoogLeNet for image visual feature extractor and an LSTM for generating image captions. Spatial-Attention takes advantage of ResNet as the visual extractor. Up-Down uses two kinds of visual features: one adopts Faster R-CNN as the powerful encoder which has been trained on the large scale Visual Genome data set, and another one uses the ResNet as the visual extractor. For fair comparison, we use the corresponding visual feature extractor of different baseline model to extract visual feature. The extracted visual features are then put into the visual encoder to compute the relationship of different objects. The preprocessed visual feature by our visual encoder is denoted as "V". We represent the de-noised textual feature as "T". For the added sum result of these two kinds of features is represented by "VT".

We can see the experiment result in Table 2 when we replace the information sources of several baseline models. Google NIC has a better performance on all the evaluation metrics except Bleu-1 by using the attended visual features "V" as the information sources. As soon as the de-noised textual features add with the attended visual features as the information sources, the model got a more powerful result, especially in CIDEr. This shows the usefulness of the similar sequences retrieved by the retrieval-based method. Spatial-Attention also achieved a higher score when the decoder takes the

attended visual feature or the added sum result as input. It shows the little helpful of the de-noised textual features. There is also an improvement on the Up-Down-ResNet model when we use both the de-noised textual feature and the original visual feature as the input of the decoder compared with the basic experiment results. Up-Down-R-CNN with the added sum result also achieves better results. Up-Down here is trained by the cross-entropy loss function.

**Table 2.** Experiment results when we replace the information sources by the visual encoder result "V" or the added sum result "VT".

| Model | B@1 | B@4 | M | R | C | S |
|---|---|---|---|---|---|---|
| Google NIC [46] | | | | | | |
| Baseline | 70.7 | 27.7 | 23.7 | 51.7 | 85.5 | 17.0 |
| w/V | 70.9 | 29.8 | 24.5 | 52.5 | 93.3 | 17.5 |
| w/VT | 71.0 | 30.1 | 25.2 | 52.7 | 95.6 | 17.9 |
| ADP-ATT [6] | | | | | | |
| Baseline | 73.4 | 30.4 | 25.7 | 54.9 | 102.9 | 18.7 |
| w/V | 73.9 | 32.9 | 26.3 | 54.8 | 105.6 | 19.2 |
| w/VT | 74.6 | 33.4 | 26.5 | 55.1 | 106.0 | 19.3 |
| Up-Down-ResNet [23] | | | | | | |
| Baseline | 73.4 | 33.4 | 26.1 | 54.4 | 105.4 | 19.2 |
| w/V | 74.2 | 33.7 | 27.1 | 55.6 | 110.5 | 20.1 |
| w/VT | 75.0 | 34.8 | 27.7 | 56.2 | 111.5 | 20.7 |
| Up-Down-R-CNN [23] | | | | | | |
| Baseline | 77.2 | 36.2 | 27.0 | 56.4 | 113.5 | 20.3 |
| w/V | 76.1 | 36.2 | 28.1 | 56.6 | 114.8 | 21.1 |
| w/VT | 76.4 | 36.4 | 28.3 | 57.0 | 115.6 | 21.2 |

As for that the deep reinforcement learning can alleviate the so-called exposure bias in text generation. We conduct experiments on Up-Down-ResNet and Up-Down-R-CNN with CIDEr optimization. Our experiment results are presented in Table 3. For the performance of the baselines, we directly report from the original papers. We can see the experiment result when the baseline takes the preprocessed visual feature "V" or the added sum result "VT" as the information sources. It is clearly to see these two kinds of information sources can bring improvements more or less on these evaluation metrics. Up-Down is a very strong baseline, our preprocessed information sources can still bring improvements under the reinforcement learning settings, proving the effectiveness of the preprocess component.

**Table 3.** Experiment results on the Up-Down model with different kind of information sources trained by the reinforcement learning method.

| Model | B@1 | B@4 | M | R | C | S |
|---|---|---|---|---|---|---|
| Up-Down-ResNet [23] | | | | | | |
| Baseline | 76.6 | 34.0 | 26.5 | 54.9 | 111.1 | 20.2 |
| w/V | 76.9 | 34.6 | 27.5 | 56.3 | 113.7 | 20.5 |
| w/VT | 77.2 | 35.2 | 28.1 | 56.4 | 114.5 | 20.9 |
| Up-Down-R-CNN [23] | | | | | | |
| Baseline | 79.8 | 36.3 | 27.7 | 56.9 | 120.1 | 21.4 |
| w/V | 78.0 | 36.7 | 28.0 | 57.0 | 121.3 | 21.5 |
| w/VT | 79.5 | 37.3 | 28.2 | 57.6 | 122.5 | 21.8 |

Ablation study of our model. It is clearly to see in Figure 1 that our decoder uses both the attended visual features "V" and the de-noised textual features "T" as the information sources. We denoted as

"V T". To verify the effectiveness of the de-noised textual features "T", we also make experiment on the decoder with only the attended visual features "V" as information sources.

The evaluated result with beam size equals 1 is shown in Table 4. It shows the decoder with "V T" acquires superior experiment results on all the evaluation metrics. This verified the effectiveness of the de-noised textual features.

**Table 4.** Experiment results of the ablation study of our model when the decoder takes the visual features "V", or takes both the visual features "V" and the textual features "T" as the information sources.

|  | Cross-Entropy Loss | | | | | | Self-Critical Loss | | | | | |
|---|---|---|---|---|---|---|---|---|---|---|---|---|
|  | B@1 | B@4 | M | R | C | S | B@1 | B@4 | M | R | C | S |
| w/V | 75.0 | 33.3 | 27.4 | 55.7 | 113.0 | 20.7 | 79.2 | 37.7 | 28.3 | 58.4 | 126.0 | 21.7 |
| w/V T | 75.5 | 33.9 | 27.8 | 56.4 | 113.9 | 21.0 | 79.7 | 38.2 | 28.5 | 58.6 | 127.1 | 21.8 |

## 5. Conclusions

In this paper, we propose a novel image captioning model which incorporates the generation-based approach and the retrieval-based method. The proposed model makes use of both the image visual features and the retrieved similar sequences textual features to enrich the information sources for captioning. We use the image-guided cross-modal attention mechanism to acquire the textual features that conditioned on the visual features to acquire the de-noised textual features. Finally, the decoder module takes both the de-noised textual feature and the attended visual feature to generate the final captions. Furthermore, the integrated visual encoder and the de-noising module can be viewed as a preprocessed component which help provide fine-grained image representations for these images to text generation task. Extensive experiments on the MSCOCO image captioning data set show that our method achieves better performance in both quantitative and qualitative evaluations.

**Author Contributions:** S.Z., L.L., H.P., Z.Y. and J.Z. conceived and designed the experiments; S.Z. performed the experiments; L.L., H.P., Z.Y. and J.Z. analyzed the data; S.Z., L.L. and H.P. wrote the paper. All authors interpreted the results and revised the paper. All authors have read and agreed to the published version of the manuscript.

**Funding:** This research was funded by the National Natural Science Foundation of China (Grant nos. 61972051, 61932005, 61771071).

**Conflicts of Interest:** The authors declare no conflicts of interest.

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
