# Peer review of "Image Caption Generation via Unified Retrieval and Generation-Based Method"

_applsci, doi:10.3390/app10186235_

Round 1
Reviewer 1 Report
The paper describes image captioning problem using combination of generation-based and retrieval-based approaches.
My comments to the authors:
- The first paragraph of the Introduction is almost precise copy of the Abstract. They must be different.
- Line 60: “shows that we our model have achieved the state-of-the-art results.”. The word “we” is odd in this sentence.
- Part “Related works”. Why didn’t you write anything about models combined retrieval-based and generation-based techniques? In my opinion it is the most important models to compare your model with. For example, with [11] (“A Unified Generation-Retrieval Framework for Image Captioning“).
- Line 93: “Different with the above mentioned methods that exploited the Transformer model.”. It is unclear what do you want to say with this sentence.
- Line 102: Why did you cite GoogLeNet but didn’t cite VGG? The commonly used encoders for image captioning task are ResNet and VGG.
- The methods with which you approach is compared are far from state-of-the-art. The most of them are dated to 2018 year. At the same time there are a number of paper whose quality is better than your model, for example:
Cornia, M.; Stefanini, M.; Baraldi, L.; Cucchiara, R. Meshed-Memory Transformer for Image Captioning. Proceedings of the IEEE/CVF Conference on Computer Vision and Pattern Recognition, 2020, pp. 10578–10587.
Huang, L.;Wang,W.; Chen, J.;Wei, X.Y. Attention on attention for image captioning. Proceedings of the IEEE International Conference on Computer Vision, 2019, pp. 4634–4643.
Pan, Y.; Yao, T.; Li, Y.; Mei, T. X-Linear Attention Networks for Image Captioning. Proceedings of the IEEE/CVF Conference on Computer Vision and Pattern Recognition, 2020, pp. 10971–10980.
Nevertheless the proposed approach is interesting but it is necessary to mention in the paper, that it doesn’t beat current state-of-the-art-models but shows promising results compared to a strong baselines. - In the Figure 2 it would be interesting not only to show retrieved caption, but also retrieved similar image.
- Line 287. Why did you use exactly these models to test your hypothesis? For example model [38] is rather old and weak.
- Line 320. “Both of them trained by the cross entropy loss.” Why you didn’t use self-critical training like you used in the first part of your work? Maybe you could achieve greater overall performance for the proposed model?
- Lines 339-341. As far as I know you shouldn’t have this section at all if you don’t have any acknowledgments.
-
The reference part should be improved. There are several cited arxiv papers have been formally published.
Author Response
Thanks you so much for your comments. I have modified our paper according to these valuable suggestions. The detailed modified place are marked as red text in the attachment.
- The same part of the abstract and the introduction is a mistake. I edit the introduction from line 16 to line 39.
- Thank you for the advice. The word “we” is redundant and was deleted.
- Thanks again. The reason I didn’t write anything about models combined retrieval-based and generation-based techniques mainly because there is only one paper [11] (“A Unified Generation-Retrieval Framework for Image Captioning“) I know that incorporate the generation-retrieval based method for image captioning. I add [11] in “Related works” in line 79 and make comparison with our methods in the experiment part in line 259.
- The sentence “that exploited the Transformer model” mainly want to say that the above mentioned model adapted from the Transformer model. But that description is unclear. Thus, I changed the original sentence “Different with the above mentioned methods that exploited the Transformer model” (original line 93) into “Different with the above mentioned methods”(now line 108).
- Yes, the commonly used encoder are ResNet and VGG. I cite VGG in line 117.
- Thanks for your advice. The Meshed-Memory Transformer,Attention on Attention model and the X-Linear Attention model got SOTA results than ours. I cited these papers in related work (line 102) and mentioned in experiment that they are superior than our model (line 264).
- Thank you so much for your advice. I edit Figure 2(now Figure 3) and add the retrieved similar image.
- I use these models to test the hypothesis is because I am familiar with these models and for the experiment convenience.
- I add the experiment results that further trained the model by the self-critical training in the paper, and the result achieve greater.
- Yes, I deleted this section.
- I edit the reference papers that cited as arxiv preprint except for [13] and [33]. I didn’t find their exactly journal information.

Reviewer 2 Report
1) Presentation of the content in Introduction and Model arch. must be improved.
Also please ensure changes in sentence formation, there were lot of sentences that were unclear. Also Figure 1 and Figure 2 image captions are the same.
Please put more figures expanding the details of your model architecture.
2) In my opinion for the sake of readability of the work:
a) You can reduce your explanation of your model architecture and just present the high-level design.
b) Present your results and emphasize the key contribution of this work: combining retrieval based and generation based features.
c) Expand the details of your model architecture after your conclusions (if it is allowed by the Journal author guidelines.)
Please see my comments in the attached pdf.

Author Response
Response to Reviewer 2 Comments
Thank you so much for your comments. I have modified our paper according to these valuable suggestions. The detailed modified place are marked as red text and the deleted text are marked as green in the pdf.
Comments in the webpage
Point 1: 1) Presentation of the content in Introduction and Model arch. must be improved.
Also please ensure changes in sentence formation, there were lot of sentences that were unclear. Also Figure 1 and Figure 2 image captions are the same. Please put more figures expanding the details of your model architecture.
Response 1: Thanks for your advices. The introduction is modified from line 16 to line 39.
I checked sentences of my paper and make some changes.
Figure 1 and Figure 2(now Figure 3) have same image captions is a mistake I’ve made before and have been corrected.
For the model architecture, I add Figure 2 to expand more details of the visual encoder and the de-noising module.
Point 2: In my opinion for the sake of readability of the work: a) You can reduce your explanation of your model architecture and just present the high-level design.
Response 2: Thanks for your suggestions. I delete some model architecture explanations. The deleted sentences are marked as green in the attachment.
Point 3: b) Present your results and emphasize the key contribution of this work: combining retrieval based and generation based features.
Response 3: Thanks again. From line 275 to line 285, I present the experiment results and emphasize the key contribution of this work is combing retrieval based and generation based features.
Point 4: c) Expand the details of your model architecture after your conclusions (if it is allowed by the Journal author guidelines.)
Response 4: Thanks for your advice. Although some detailed model architecture introduction is deleted, the whole model architecture is still clearly presented. Thus, I think it may be unnecessary to expand the details of our model after conclusions.
Point 1: In line 38, Unclear? Please clarify - whole image scene, or give a reference.
Response 1: Whole image scene here means the whole image. This means the model analyse the whole image each time when generate a word, not focus on a specific object in the image like attention when generate a word.
Point 2: always lack? - a harsher claim Suggestion: the generated captions mainly suffer from a lack of fluency, diversity ....
Response 2: Thanks for your advice, I changed the “always lack” to “mainly suffer from”.
Point 3: line 49, of these?? Please take a look at the sentence. I understand what you
wish to say, but it might be unclear to some readers.
Response 3: I changed the original sentence to “we fully explore potential hidden correlations of different image regions to form a spatial and relational representation.”
Point 4: Suggestion: You can itemize the (a), (b), (c) for better readability
Response 4: Thanks again, I itemized the (a), (b), (c).
Point 5: In Figure 1: The caption
Point 5: In Figure 1: The caption doesn't match the image??? Is this your model arch diagram- retrieval and generation based method?
Response 5: The retrieved similar image may match the input image or not. Here I list an image that match the input image in Figure 1. If the retrieved similar image is not that match the input image, the de-noising module can de-noise the retrieved similar sequences according to the attended visual features, and the decoder can generate descriptions according to the attended visual features and the de-noised textual features. If the retrieved similar image is quite similar with the input image, the similar sequences can bring a set of useful semantic information for our model.
Point 6: Line 89, A suggestion: So when using specific keywords such as Transformer it would be nice to emphasize it using italic fonts. This lets the reader know that it is a special term and is nit a transformer with a capital T.
Response 6: Thanks, I already edit Transformer into italic fonts.
Point 7: Line 95: Could help us? Means it may or may not? You already know the answer, as this is the research you have presented - you can just make a claim and say we prove it does help filter retrieved similar sequences...
Response 7: Yes, the model can filter retrieved similar sequences. I changed the word “could” into “does”.
Point 8: Line 114: also tell what is m and d-model.
Response 8: m is the number of objects. d_model is the latent dimension.
Point 9: Line 132: "form inner groups that are of importance to our approach"
Response 9: I modified the sentence “form an inner groups we cared about” to “form inner groups that are of importance to our approach” in line 146.
Point 10: Line 143: Nice!!! Suggestion: You can break your original network Figure 1 into some detailed sub-figures as you explain your network.
Response 10: Thanks for your suggestion, I add a sub-figure (Figure 2) to introduce the architecture of the visual encoder and the de-noising module.
Point 11: Line 162: as you have already implemented this, A suggestion to avoid using could or may instead of "we could get" we get a set of textual. We intend to extract a set of textual. (This suggestion is for the entire section describing the model architecture)
Response 11: Yes, I will avoid to use could or may and changed in the paper.
Point 12: Line 174: As Figure 1 needs to be referred again and again, I would recommend putting an image of your decoder in the model arch here for ease of readability.
Response 12: As the detailed decoder architecture is clearly shown in Figure 1. I add Figure 2 to introduce the visual encoder and the de-noising module.
Point 13: Sentence starting with And? No need for "And we have:", We adopt the self-criti.......sampled sequences, which is:
Response 13: Thanks for you advice, I edit this sentence into “We adopt the self-critical sequence training method~\cite {rennie2017self} on these sampled sequences which is:” in line 196.
Point 14: As is shown in Figure 2, we show to In Figure 2 we show
Response 14: Yes, I changed the original sentence into “In Figure 3 we show” in line 275.

Round 2
Reviewer 1 Report
I agree with the answers and I think the article can be published.